

# Application of machine learning to forecast agricultural drought impacts for large scale sub-seasonal drought monitoring in Brazil

Joseph W Gallear[1], Marcelo Valadares Galdos[1], Marcelo Zeri[2], and Andrew Hartley[3]

[1]Rothamsted Research, West common, Harpenden, UK
[2]National Center for Monitoring and Early Warning of Natural Disasters (Cemaden), São José dos Campos, Brazil
[3]Met Office Hadley Centre, FitzRoy Road, Exeter, UK

**Correspondence:** Joseph W Gallear (joe.gallear@rothamsted.ac.uk)

**Abstract.** Drought events have increased in frequency and severity in recent years, and result in significant economic losses. Although the Brazilian semi-arid northeast has been historically associated with the impacts of drought, drought is of national concern, from 2011-2019, drought events were recorded in all Brazilian territories. Droughts can have major consequences for agricultural production, which is of particular concern given the importance of soybeans for socio-economic development. Due
to its regional heterogeneity, it is important to develop accurate drought forecast and assessment tools for Brazil. We explore machine learning as a method to forecast the vegetation health index (VHI), for large scale monthly drought monitoring across agricultural land in Brazil. Furthermore, we also determine spatio-temporal drivers of VHI across the wide variation in climates, as well as evaluate machine learning performance for El Niño-Southern Oscillation (ENSO) variation, forecasting of the onset of drought impact, and how the trade off between spatial variation and sample size affects model performance. We show that
machine learning methods such as gradient boosting methods are able to more easily forecast vegetation health in the north and north east Brazil than south Brazil, and perform better during La Niña events than El Niño events. Drought impacts which reduce VHI below the typically used 40% threshold can be forecast across Brazil with similar model performance. SPEI is shown to be a useful indicator of drought impact, with 3 month accumulation periods preferred over 1 and 2 months. Results aim to inform future developments in operational drought monitoring at the National Center for Monitoring and Early Warning
of Natural Disasters in Brazil (CEMADEN). Future work should build upon methods discussed here to improve drought forecasts for agricultural drought response and adaptation.

## 1   Introduction

Drought events have increased in frequency and severity in recent years and can result in significant economic losses (Cunha et al., 2019; Herweijer and Seager, 2008; Marengo et al., 2017; Brito et al., 2018). Droughts are an extended period in which a
water deficit occurs, usually because precipitation is less than average resulting in water scarcity (Cunha et al., 2019). Typically, drought is a slow onset hazard, the effects of which may take many months before the propagation of the deficiency of rainfall can be detected (Cunha et al., 2019; Mo, 2011). Droughts can have significant consequences for sectors including drinking water supply, crop production, waterborne transportation and electricity production (Van Loon, 2015). Drought indices such as the Standardized Precipitation Index (SPI) are commonly used tools to assess drought impacts. SPI is a standardized measure





of the probability of a particular deficit of rainfall over a given period of months. SPI is a useful index as it can compare and describe drought events for different regions and timescales. For this reason SPI has widespread applicability for identifying drought events (Dai et al., 2020; Angelidis et al., 2012).

Agricultural drought can significantly reduce crop yields, with subsequent economic losses. Agricultural drought reflects the extent to which soil moisture is lower than the required amounts for plant growth by accounting for soil moisture and 30 plant characteristics (Liu et al., 2016). Sensitivity to drought effects can depend on management factors such as crop selection, irrigation, and tillage practice, as well as climate variability (Wilhelmi and Wilhite, 2002). Agricultural drought has been effectively detected using the vegetation health index (VHI), a proxy for the estimation of vegetation health (Kogan, 2002; Wu et al., 2020). This is because VHI derived from AVHRR (Advanced very high resolution radiometer) data responds cumulatively and quickly to changes in vegetation greenness, and so the effect of drought can be measured much earlier than that derived from 35 weather data or other drought monitoring tools (Kogan, 2002).

Drought monitoring using vegetation indices such as VHI or NDVI (Normalized difference vegetation index) or VCI (Vegetation condition index) has been developed in several locations using satellite imagery from products such as MODIS, and NOAA STAR (Sadiq et al., 2023; Kloos et al., 2021). Machine learning has been used to forecast vegetation indices at timescales including daily, 5 and 7 day intervals (Kartal et al., 2024; Kladny et al., 2024; Reddy and Prasad, 2018), monthly 40 intervals (Lees et al., 2022), weekly timescales (Barrett et al., 2020), and average vegetation condition values aggregated over 1-3 months (Adede et al., 2019). Models used to predict VHI range from neural networks (Adede et al., 2019; Kladny et al., 2024; Lees et al., 2022; Reddy and Prasad, 2018) to ensemble tree methods such as random forest, and gradient boosting methods (Nay et al., 2018; Tanguy et al., 2023), with some studies using other methods such as gaussian process modelling (Barrett et al., 2020). Other related work has investigated the relationships between drought indicators such as SPI and SPEI 45 (standardized precipitation-evapotranspiration index) with vegetation indices (VCI & VHI) for crop-masked agricultural areas. Tanguy et al. (2023) have found that SPI and SPEI can have high correlations with VCI for relatively short accumulation periods during the dry season in Thailand. The same study also highlighted that SPEI is generally more highly correlated with crop production than SPI showing a link between evaporative demand and impact on crops.

In Brazil, drought is of high impact and importance, making up around half of natural disaster related impacts in terms of 50 number of people affected (Sena et al., 2014). Droughts are of particular concern in the northeast semi-arid region, one of the most densely populated semi-arid regions in the world, which also has the most people living in poverty in Brazil (Cunha et al., 2019; Marengo et al., 2022). Much work has focused on drought trends in Brazil, with particular focus on the north east semi-arid region (Cunha et al., 2019; Marengo et al., 2017, 2022; Rossato et al., 2017; Zeri et al., 2018). However, in recent years, drought impacts have affected all regions in Brazil (Cunha et al., 2019). Due to its regional heterogeneity, it is important 55 to develop accurate drought forecast and assessment tools for all of Brazil (Cunha et al., 2019).

Drought monitoring and dissemination of drought warnings and intensification in Brazil is undertaken by the National Center for Monitoring and Early Warning of Natural Disasters (CEMADEN). The drought monitoring system used by CEMADEN considers several drought indices including the Standardized Precipitation Index (SPI), Root Zone Soil Moisture (RZSM) from remote sensing, and vegetation indices based on remote sensing such as the vegetation health index (VHI). The SPI is computed





by making use of a long-term national database of rain gauges, which is updated daily. SPI is generated in time scales of 1 to 6 months, for agricultural drought, and in longer time scales (12 to 24 months) for hydrological drought monitoring. The RZSM is obtained from the following NASA's satellites: Gravity Recovery and Climate Experiment (GRACE; 2002-2017) and GRACE Follow On (GRACE-FO; 2018-present). The VHI is obtained from NOAA's satellites, with products released every 7 days. At CEMADEN, these variables are part of an Integrated Drought Index (IDI), which takes into account classified

versions of these products, harmonized to a common spatial resolution and domain. The IDI is then used to make the diagnostic of current drought conditions over all regions of the country. The index is made public in reports and monthly meetings with stakeholders, which are also open to the general public.

In this study, we aim to build on drought monitoring work in Brazil by assessing the potential for machine learning based operational drought impact forecasting using satellite based VHI observations and drought indices at large scale across Brazil.

This work aims to benefit stakeholders at CEMADEN to inform drought monitoring efforts for agricultural impacts. Drought forecasts are made using previous variation in vegetation health index as well as the impact of meteorological drought indicators such as the Standardized precipitation index (SPI), Standardized precipitation-evapotranspiration index (SPEI), and root zone soil moisture (RZSM). VHI is an important indicator of how drought conditions may affect crops in agricultural areas. VHI values below 40% typically indicate drought conditions. Analysis in this study compares a range of machine learning methods

for drought forecasting in section 3.3. Furthermore, the potential predictability of indicators through the impact of temporal and spatial variability on forecasting model performance is determined in sections 3.1 and 3.7. Finally, the most important indicators, both overall and spatially for VHI forecasts are analysed in section 3.6. This approach is designed to show the potential for machine learning methods for operational forecasting of VHI across Brazil at monthly timescales.

This study is the first to forecast drought impacts on vegetation health using machine learning in Brazil at sub-seasonal

timescales relevant to agricultural adaptation. This is also the first study to describe spatio-temporal relationships between drought indicators SPEI, SPI, root zone soil moisture, and vegetation health index across agricultural areas in Brazil.

## 2 Methods and data

As a scoping exercise for building a drought forecasting machine learning method for Brazil, different methods and indices are evaluated here across different spatial scales across the country. Firstly, various machine learning methods described in

section 2.4 are compared across randomized evaluation years (section 3.3. Secondly, drought indices described in section 2.2 are compared and evaluated against their correlation with the vegetation health index (section 3.2), model performance, and contribution towards model forecasts (section 3.6). Thirdly, through the clustering of VHI data described in section 2.7, the spatial scale at which to make distinctions between climate zones is compared (section 3.7). The methods in this study are kept broad to enable others wishing to learn from this work and build their own machine learning forecasting pipelines for other

countries and regions at large scales using satellite and remote sensing data products.

Following is a description of the study area of focus for this paper. We use satellite data to enable large scale drought forecasts across major agricultural regions in Brazil. The study area contains a wide variation in climate (Beck et al., 2018). Notably, the

hot semi-arid northeast, in contrast to the humid sub-tropical climate in the south and tropical savanna in the central region, as described by the Köppen-Geiger climate classifications (Beck et al., 2018; Peel et al., 2007). Data was obtained at different

spatial and temporal scales, some of which required processing to convert data types to a common spatial and temporal scale. The common spatial scale chosen was 0.25°, with a monthly time step. This was chosen to minimise the amount of spatial up-scaling and down-scaling required.

## 2.1   Study area

To provide estimates of drought impacts across a range of environments gridded vegetation health index data derived from

NOAA (National Oceanic and Atmospheric Administration) STAR satellite (Kogan, 1997; Kogan et al., 2013) was obtained for the entirety of Brazil. This data is then filtered using harvested areas from the crop grids dataset (Tang et al., 2023). The data was filtered to only contain grid cells which are above the $75^{th}$ percentile of harvested area across the distribution of harvested area in Brazil. This is to ensure that the grid cells used for training and evaluation are most likely to be indicative of cropland for two major crops grown in Brazil, soybean and maize. Choosing maize and soybean growing areas provides a

large spread across different climatic zones of Brazil, and ensures representation of two crops with economic and food security value. Figure 1 shows the spatial distribution of soybean (a) and maize (b) growing areas.

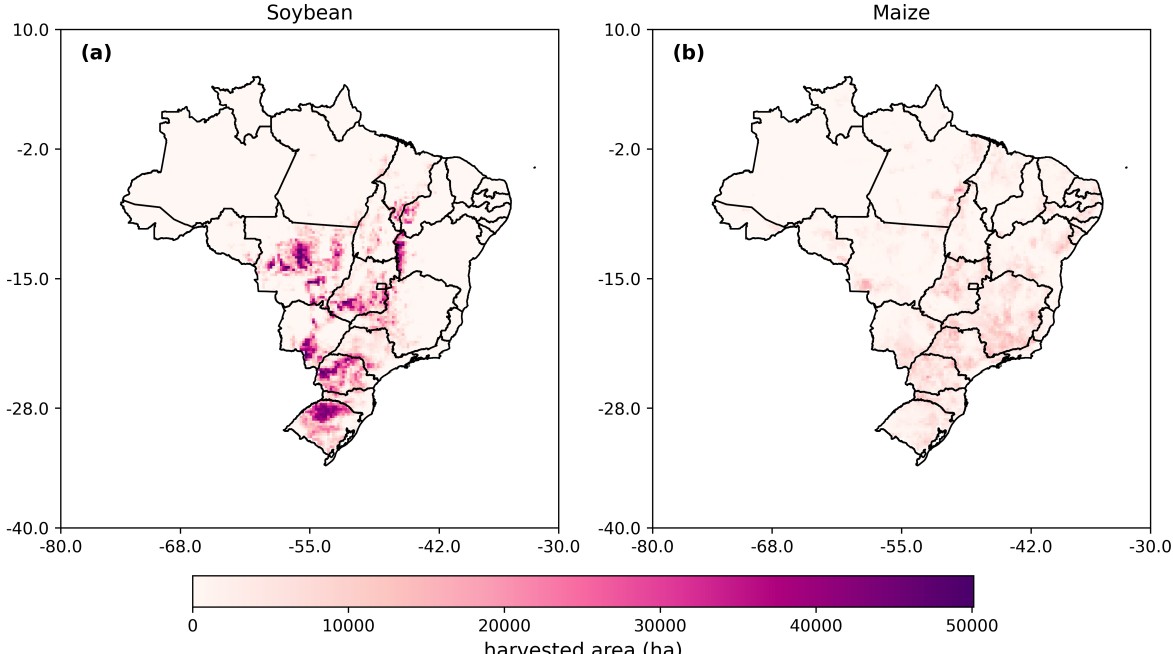

**Figure 1.** Maps of harvested areas (circa 2020) across Brazil taken from the Crop grids database. Panel (a) shows soybean harvested area. Panel (b) shows maize harvested area.





Using the approach here to select for regions, the area under cropland was obtained for a range of locations across Brazil. Much of the most intensely farmed soybean area is in the state of Mato Grosso in central Brazil, as well as Rio Grande do Sul and Paraná in the South and some locations in Bahia in the north east. Maize is a lot less intensely farmed than soybean but just as widespread throughout the country. More maize is grown in Minas Gerais than soybean and is more widespread in the north east of the country.

The large spatial scale of this work makes model training particularly challenging. Agricultural land in Brazil is made up of multiple biomes of differing soil moisture, rainfall, and temperature characteristics (Cunha et al., 2019; Lopes Ribeiro et al., 2021). Meteorological events such as ENSO also affect different parts of the country in different ways. Typically, during El Nino events, there is a reduction in precipitation in the north and northeast regions, while the south experiences higher frequency of heavier rains. In La Nina events, the situation is reversed, with the north and northeast experiencing greater than average rainfall, and south subject to more severe droughts (Cirino et al., 2015).

## 2.2 Drought Indices

Drought indices were taken from a range of sources, re-sampled to ensure consistent spatial and temporal resolution (see section 2.3) and then assimilated to create a combined data set to describe drought conditions across Brazil at 0.25° spatial resolution for each month. To obtain consistent dates across data sources, the data set ranges from 2003 to 2022. These years also account for seasonal variability and cyclical climate processes, including ENSO which is addressed in this study.

### 2.2.1 Vegetation Health Index (VHI)

The vegetation health index is a proxy for estimating overall vegetation health, and is an index expressed in percentage. VHI values below 40% indicate stress conditions. VHI is a composite index which is comprised of the vegetation condition index (VCI) and temperature condition index (TCI). VHI is obtained through the following formula:

$$VHI = \alpha * VCI + (1 - \alpha) * TCI \tag{1}$$

Where $\alpha$ is a coefficient used to determine the relative contribution of TCI and VCI to VHI. The vegetation health index data was obtained from the NOAA STAR satellite based vegetation health system. The NOAA STAR system uses data and products from GOES (Geostationary Operational Environmental Satellite), METEOSAT, MTSAT, and DMSP. Satellite observations are primarily based on radiance measurements in the visible, near infrared, and 10.3-11.3 micrometers thermal bands taken by the Advance Very High Resolution Radiometer (AVHRR) found on NOAA polar orbiting satellites. The visible and infrared observations are used to determine NDVI which is in turn converted to TCI, VCI and the vegetation health index (VHI) after filtering out noise and adjusting for non-uniformity of the land surface due to climate and ecosystem differences (Kogan, 1997). VHI data was obtained at the resolution of 0.036° (4km) but then up-scaled to 0.25° to bring to the common spatial resolution of the majority of the input data.



### 2.2.2 Soil Moisture

Soil moisture is essential to measure the propagation of meteorological drought into agricultural drought and water stress in plants (Zeri et al., 2018, 2022). In this work, soil moisture was obtained from the NASA GRACE satellite (Li et al., 2019). The NASA GRACE satellite data is based on 2 satellites which record changes in the earths gravity field caused by the redistribution of water. The original output from the satellites does not distinguish between surface soil moisture, root zone soil moisture and groundwater storage. Therefore, in order to isolate each of the groundwater components from total soil water storage, the satellite data is integrated with other ground and space based meteorological data within the catchment land surface model using ensemble Kalman smoother type data assimilation (Li et al., 2019). Root zone soil moisture was obtained from GRACE for 0.25° grid scale, and a weekly timescale. Temporal resolution was reduced to monthly by averaging soil moisture percentage across 4 week intervals.

### 2.2.3 Standardized precipitation index (SPI)

The standardized precipitation index (SPI) is a drought index with wide comparability for different locations due to spatially consistent standardization. This makes the SPI a useful index for constructing a model of drought propagation across such a wide spatial domain as agricultural land in Brazil. SPI was first proposed by McKee et al. (1993) to quantify the probability of occurrence of a precipitation deficit at a particular monthly timescale. To determine SPI, precipitation data are fitted to a probability distribution function (usually either gamma or Pearson), before the inverse normal distribution function is used to re-scale probability values, leading to SPI values with a mean of zero and standard deviation of one (Cunha et al., 2019). SPI is calculated over different monthly timescales, Here we use 1,2 and 3 month SPI. Various studies have shown that SPI-3 has the strongest correlation with vegetation response (Sepulcre-Canto et al., 2012) however we also assess 1 and 2 month accumulation periods, which may also be useful for dry environments (Tanguy et al., 2023). SPI is a widely used index recommended by the world meteorological organisation (WMO). It is also used for operational drought monitoring at CEMADEN (Cunha et al., 2019).

SPI data was taken from the NOAA NIDIS Global precipitation climatology centre (GPCC) (Ziese et al., 2011). We selected SPI data fit to a Gamma distribution. Data was obtained at a 1° resolution then up-sampled using a k-nearest neighbours algorithm to obtain a consistent spatial resolution with the rest of the data set at 0.25°.

### 2.2.4 Standardized Precipitation-Evapotranspiration Index (SPEI)

SPEI (Standardized Precipitation-Evapotranspiration Index) is based on the calculation of SPI, however SPEI is determined by computing a climatic water balance, then using this metric to determine probability of a water balance deficit for a given period of time. The climatic water balance is given by calculating the difference between precipitation and potential evapotranspiration (PET). Similar to SPI, a statistical distribution is then used to fit the data, and the data is standardized to produce a mean of zero and standard deviation of one (Beguería et al., 2014).





SPEI data was taken from the global SPEI database (SPEIbase) which was originally at a 0.5° spatial resolution (Beguería et al., 2014). Beguería et al. (2014) use a log logistic distribution to fit the SPEI index. SPEI is an advancement upon the SPI (standardized precipitation index) because the incorporation of evapotranspiration effects accounts for temperature effects on drought which have been shown to significantly affect drought conditions (Rebetez et al., 2006). SPEI values are determined for a number of months, termed accumulation periods. Different accumulation periods could be more useful for specific representations (e.g. longer accumulation periods could be more correlated with longer term storage effects such as groundwater). Here we assess 1-3 month accumulation periods for consistency with SPI accumulation periods.

### 2.2.5 ERA5 Reanalysis predictors

Further data was obtained from the monthly averaged ERA5 database (Hersbach et al., 2019). ERA5 is a reanalysis database which combines models and observations using data assimilation to provide better estimates of meteorological variables (Hersbach et al., 2019). Although ERA5 has an hourly global coverage, we use monthly averaged estimates to allow for consistency with the rest of the data used for this study.

From this resource, 2 metre temperature, potential evaporation, and surface thermal (longwave) radiation downward were all obtained. Temperature variables are important to capture drought effects brought on by high temperatures rather than solely a deficit in rainfall. This can be especially important for flash drought events, which are typically caused by compounding effects of rainfall deficits and high temperatures which increase evaporative stress (Christian et al., 2021).

### 2.2.6 Precipitation

Precipitation data was obtained from the Climate Hazards Group Infrared Precipitation with Station data (CHIRPS) database (Funk et al., 2015). CHIRPS is a quasi-global database (ranging from 50°N - 50°S) which is available at multiple spatial resolutions including 0.25°. The CHIRPS dataset combines satellite data with *in situ* measurements to provide a gridded dataset of appropriate spatial extent for this study. CHIRPS has been validated against other datasets and *in situ* observations and has been used for similar studies in other regions (Lees et al., 2022).

### 2.3 Data Processing and Sampling Methods

Data was originally obtained at a range of different spatial and temporal resolutions. Table 1 shows the original resolution of each of the indices used. Where spatial resolution has been decreased (spatial down sampling) this is done through averaging. Where spatial resolution has increased (spatial up sampling) this is calculated using a k-nearest neighbours algorithm. All data was spatially corrected to a 0.25° spatial resolution. Some data was obtained at weekly or daily timescales, in this case, data was averaged per month for each grid cell location to obtain average monthly estimates of each variable.





**Table 1.**

Predictor variables considered for use in this study with original spatial (degrees) and temporal resolution, source, and abbreviation used in this paper.

| Predictor | Spatial resolution | Temporal resolution | Source | Abbreviation |
|---|---|---|---|---|
| 2 Metre temperature | 0.25 | Monthly | ERA 5 | t2m |
| Potential evaporation | 0.25 | Monthly | ERA 5 | pev |
| Surface thermal radiation downwards | 0.25 | Monthly | ERA 5 | longrad |
| Root zone soil moisture | 0.25 | Weekly | NASA GRACE | RZSM |
| Precipitation | 0.25 | Daily | CHIRPS | precip |
| SPEI 1 | 0.50 | Monthly | LCSC | SPEI1 |
| SPEI 2 | 0.50 | Monthly | LCSC | SPEI2 |
| SPEI 3 | 0.50 | Monthly | LCSC | SPEI3 |
| SPI 1 | 1.00 | Monthly | GPCC | SPI1 |
| SPI 2 | 1.00 | Monthly | GPCC | SPI2 |
| SPI 3 | 1.00 | Monthly | GPCC | SPI3 |

## 2.4 Forecasting Methods

We evaluate a range of machine learning methods for the forecasting of vegetation health index 1 month in advance before using the best model to assess the performance of further forecasts aimed at predicting vegetation health index 2 and 3 months in advance. Methods here compared are Random Forest (Breiman, 2001), gradient boosting (Friedman, 2001), artificial neural networks (LeCun et al., 2015) k-nearest neighbours regression, ridge regression and a multiple linear regression for comparison.

Random forest and gradient boosting are tree based methods which construct an ensemble of decision trees. Decision trees partition data into subsets based on conditions at each leaf node of the tree. Tree depth and complexity can be specified by the user. random forest constructs a specified number of trees and then averages the result of each individual tree. Different trees are trained on different randomized sub samples of the dataset, a method known as bootstrapping. Gradient boosting methods differ from random forest in that decision trees are trained sequentially rather than simultaneously, with residual error from previous decision trees used to improve each subsequent model.

Artificial neural networks are layered networks of inter connected units which each contain a set of weights. Weights are optimized against an error term and the training data using a separate optimization algorithm. Deep neural networks are those which contain many subsequent processing layers (LeCun et al., 2015). Neural networks are flexible architectures, with many adaptations being constructed for different tasks. Here, we compare a fully connected neural network. Fully connected neural networks are named as such because each node in the preceding layer is connected to each node in the subsequent layer.





K-nearest neighbours regression is a semi-supervised learning method which uses a user defined k value to learn the k-nearest data based on a distance calculation. Most commonly this method uses the euclidean distance metric for this approach, however other distance metrics may be used (Chomboon et al., 2015). Multiple linear regression and ridge regression are used as linear comparisons to more complex methods used here to assess the appropriate level of complexity for the model required.

## 2.5 Cross validation

We cross validated models across a large span of years to provide a general picture of model performance regardless of evaluation period. Evaluation was split by year to avoid the influence of spatial autocorrelation on data leakage between training, validation and testing splits. Model evaluation metrics were obtained by training 10 separate models with the same set of hyperparameters each tested using a randomized hold out test year. Results for each model are then aggregated to produce metrics across the 10 year aggregation period. Further to this, we also evaluated optimal hyperparameter values, the results of which can be found in appendix A. To optimize hyperparameters, the data was again split by year to avoid any shared information between splits, however data was also split 3 times into training, validation and testing. For this method, we used a hold out evaluation data set of 5 random years. These years were chosen as 2006, 2011, 2016, and 2019. The rest of the data is split between 2 randomized folds based on year. The best set of hyperparameters across both folds are used to train each model before subsequent testing on the evaluation dataset. The decision was made to train and evaluate on as much data as possible here with sub-optimal parameters to provide the best indication of general model performance across a wider range of evaluation years. This allows us to better look into effects of ENSO on model performance and inter-annual variability regardless of specific hyperparameter optimization. Although hyperparameter optimization did change the results of individual models slightly, the best model was the same across both training and evaluation procedures. Furthermore, it was found that the best model results were achieved by simply training on more data, rather than a specific set of hyperparameters.

## 2.6 model evaluation methods

Model performance is evaluated using complementary mean absolute error and coefficient of determination metrics. Coefficient of determination is used to determine the performance of the model against a wide degree of variability, with high coefficient of determination indicating that the model captures both extremes at the low and high end of the distribution.

Furthermore, prediction of the onset of drought impact is evaluated in section 3.5. Here, we use precision and recall as evaluation metrics for evaluating whether the model correctly predicts when VHI decreases below 40%. The 40% threshold is chosen because it is used in many drought monitoring systems as the critical threshold at which warnings are issued (Kogan, 1997; Kogan et al., 2013; Gidey et al., 2018). Recall and precision are defined by the four classification metrics used to determine classification performance. True positives (TPs), True negatives (TN), False positives (FP), and False negatives (FN). A true positive is determined when the observed value of VHI falls below 40% and the model correctly forecasts a value of VHI below 40% for that month. Conversely, if the observed value of VHI falls below 40% but the model forecasts a value above 40% this is a false negative. Likewise, if the model forecasts a value below 40% which was not observed, this is classed





as a false positive. Finally, if both the observed and predicted values fall above the 40% threshold a true negative is determined.

Table 2 defines each of the classification values.

**Table 2.**

Definitions of classification metrics used to determine model performance for accurately predicting the onset of drought impacts on VHI.

| classification | observed value | forecast value |
| --- | --- | --- |
| True positives (TP) | $VHI < 40\%$ | $VHI < 40$ |
| False positives (FP) | $VHI > 40\%$ | $VHI < 40$ |
| False negatives (FN) | $VHI < 40\%$ | $VHI > 40$ |
| True negatives (TN) | $VHI > 40\%$ | $VHI > 40$ |

Recall and precision are defined using the classification determined in Table 2. Recall is a measure of the number of true positives as a ratio of the number of true positives plus the number of false negatives. Formally, recall is defined as:

$$recall = \frac{TP}{TP + FN} \tag{2}$$

In this manner, recall can be thought of as the performance of the model in proportion to the bias towards predicting the
250 negative class (values above 40%). Precision is similarly defined as:

$$Precision = \frac{TP}{TP + FP} \tag{3}$$

Precision is therefore defined as the number of true positives as a ratio of the number of true positives plus the number of false positives. It can therefore be thought of as the performance of the model in proportion to the bias towards predicting the positive class (values below 40%).

**2.7 Spatial clustering**

Here we use K-means clustering to define a series of training data sets based on spatial variability of VHI within each cluster. In doing so, we will answer the question of how broad a spatial domain is most appropriate for building forecast models for VHI in Brazil given the large variation in climates across the country. The number of clusters used to split the data is reduced from 10 clusters to 1, with average spatial variability within each cluster plotted against number of clusters in Figure 18. Average
spatial variability is then plotted against average model performance in Figure 19.

To do this analysis, the training and testing split used was described in section 2.5. For this analysis we used a five year hold out evaluation period, with years chosen randomly from the dataset, the rest of the data was used for training. With five complete years of monthly data, this was decided to be sufficient to assess the effect of spatial clustering on model performance.





# 3 Results

The results of this paper aim to present a first look at the potential of machine learning to produce monthly VHI forecasts and the impacts of drought on VHI across Brazil. Model performance indicates great benefit can be obtained from forecasting sub-seasonal vegetation health 1 month in advance. Forecasts further in advance, for 2 and 3 months may also be achievable but show much greater model uncertainty with methods tested.

## 3.1 Temporal autocorrelations & variability

VHI has great potential to be forecast at sub seasonal timescales due to high temporal autocorrelations in many locations, particularly the northeast. Figure 2 show a high correlation between VHI between subsequent months. High autocorrelation between monthly RZSM also indicates potential for improved forecasts. SPEI and SPI autocorrelations, as expected, depend on the accumulation period used to calculate SPI/SPEI. Precipitation autocorrelation is significantly different in southern Brazil, which indicates high inter-month variability which is confirmed in Figure 3 which displays the spatial distribution of coefficient

of variance across Brazil. Coefficient of variance is simply the standard deviation of VHI at each grid cell location divided by the historical mean at the same location. Most interestingly, northeast Brazil shows the highest coefficient of variance in VHI. The northeast also has the highest variation in rainfall, but lowest average monthly total rainfall.



**Figure 2.** 1 month temporal autocorrelations across the study area for each variable used as a predictor. Each panel shows the correlation between the each variable, and the same variable for the previous month.


Variance and autocorrelation are the temporal component of data which will affect potential machine learning forecast model performance. However, for large area prediction, spatial variation also affects the ability of models to generalize temporal trends

and so capture realistic temporal variability over a larger area. Effects of spatial variance are determined in section 3.7.

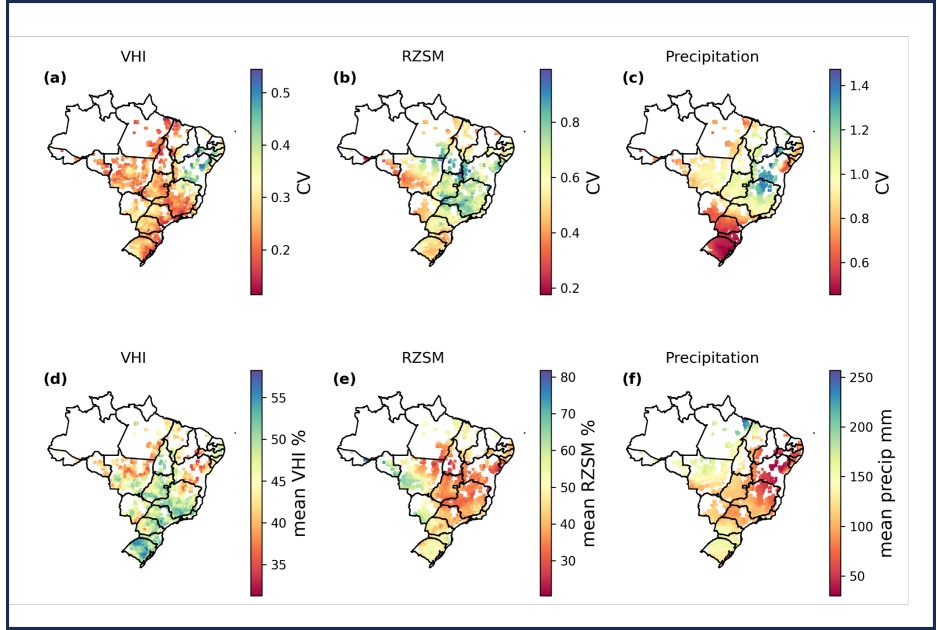

**Figure 3.** Coefficient of variance (CV) and mean VHI, RZSM, and precipitation across the spatial domain of the study. Coefficient of variance is defined as the standard deviation of monthly VHI divided by the historical mean for each location. Higher values of CV indicate larger monthly variability relative to the long term mean.

## 3.2 Drivers of VHI variability

Figure 4 shows the correlations between SPI and SPEI with vegetation health index of the following month. Longer accumulation periods lead to greater correlations with VHI. SPEI values are more strongly correlated with VHI values in some regions than SPI. Regions where this occurs include northern Mato Grosso in central Brazil, and the south. Neither SPEI or SPI have

285 very strong correlations with next months VHI in these regions, but SPEI typically has correlations which are less weak.

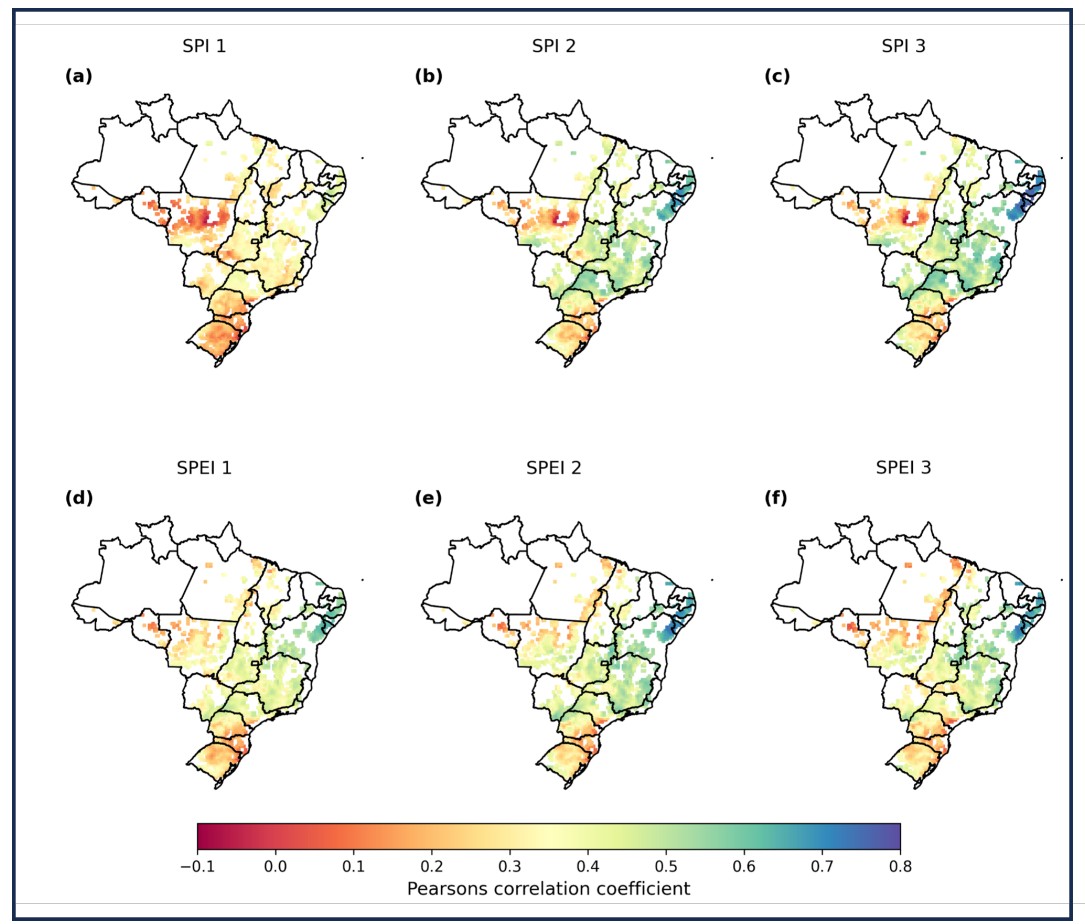

**Figure 4.** Correlation coefficient between the vegetation health index and drought indices SPI and SPEI with 1-3 month accumulation periods.

Other variables included in the modelling process may also be significant drivers of VHI. Figure 5 shows the correlations between next month's VHI and Root zone soil moisture (RZSM), precipitation, potential evaporation, downward longwave radiation, 2 metre temperature and VHI of the present month. As expected, the highest correlations are between the present and subsequent months VHI. RZSM has high correlations in the north east although very weak correlations around central Brazil. 2 metre temperature generally has greater correlations with next months VHI than downward longwave radiation.


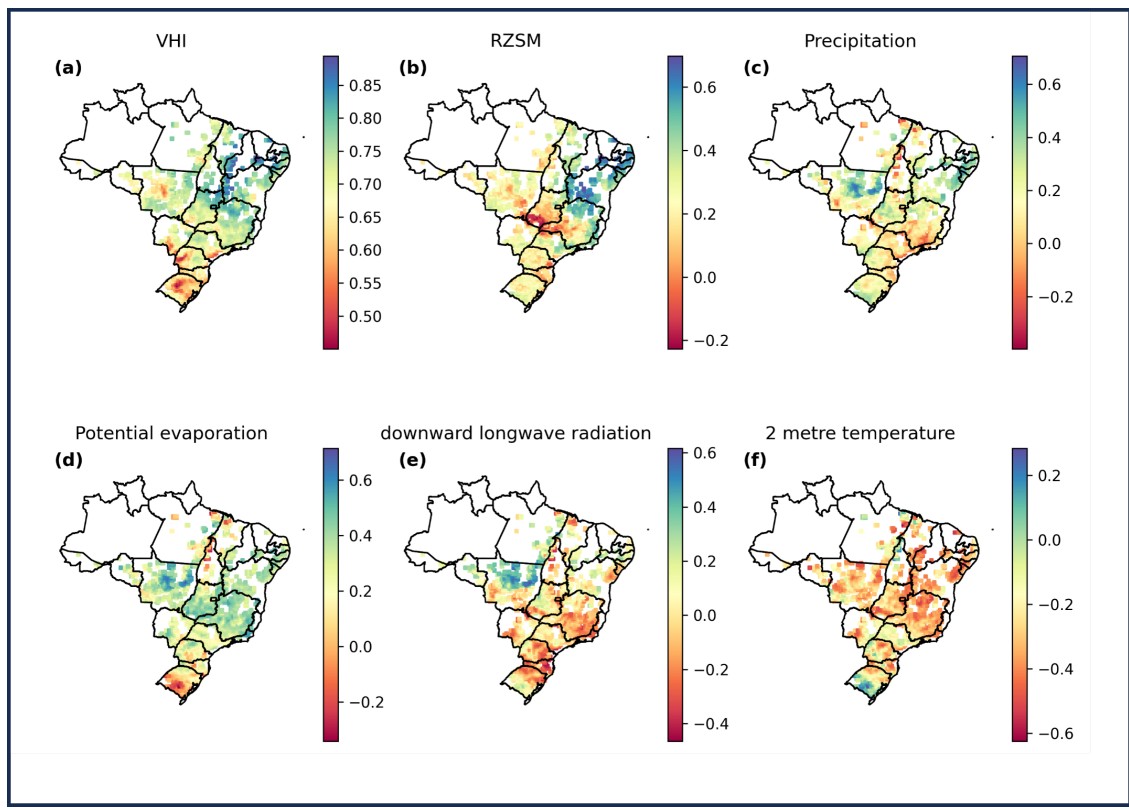

**Figure 5.** Correlation coefficient between the vegetation health index and drought indices temperature related variables, precipitation and vegetation health index of the previous month.

The correlations in Figures 4 and 5 were used to inform 1 month forecasts of VHI using a variety of machine learning methods. The results of which are presented in the following sections.

### 3.3 VHI forecasts

The initial selection of models described in section 2.4 are compared here in Figure 6. Gradient boosting model (GBM) was
generally able to achieve more consistent performance across randomized test years than the other models. For this reason, the GBM model was then chosen for further analysis including testing against months with greater lag times. SEA AV denotes a 'seasonal average' benchmark, which is simply a model which predicts each month at each location as the historical average for the month, for the location to be predicted. All models outperform this low benchmark. This allows for the conclusion that all models show performance greater than that which can be inferred entirely from seasonal variability in VHI.


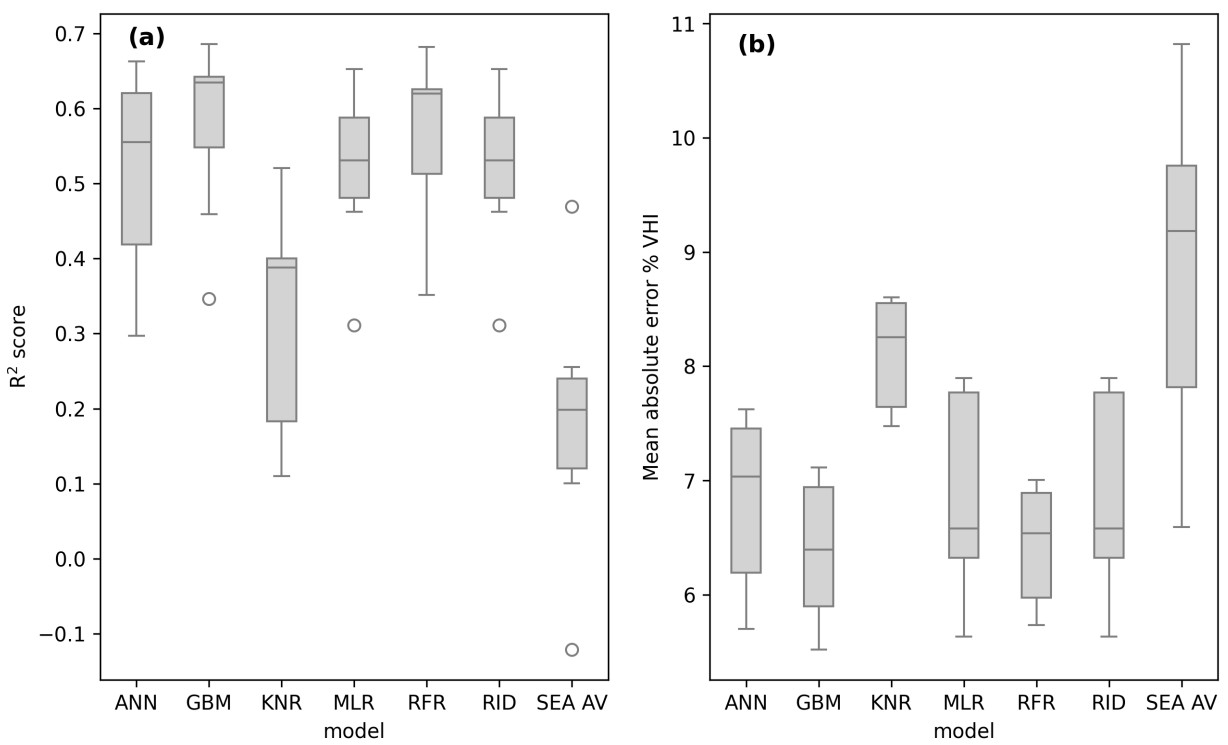

**Figure 6.** VHI forecasting model performance across a range of initially selected models. SEA AV refers to the monthly average model.

The best model from the initial comparison was taken and further assessed across the spatial domain (Figure 7 and for the mode of the southern oscillation index (SOI), Figure 10. Model performance in terms of coefficient of determination is greatest in the east whilst some of the weakest correlations are in the west of the country, south and central regions. Panel (c) of Figure 7 shows that generally $R^2$ values are between 0.6 and 0.75 for the gradient boosting machine learning model across grid cells. The distribution of mean absolute error values is less skewed, with most falling between 5-6%.


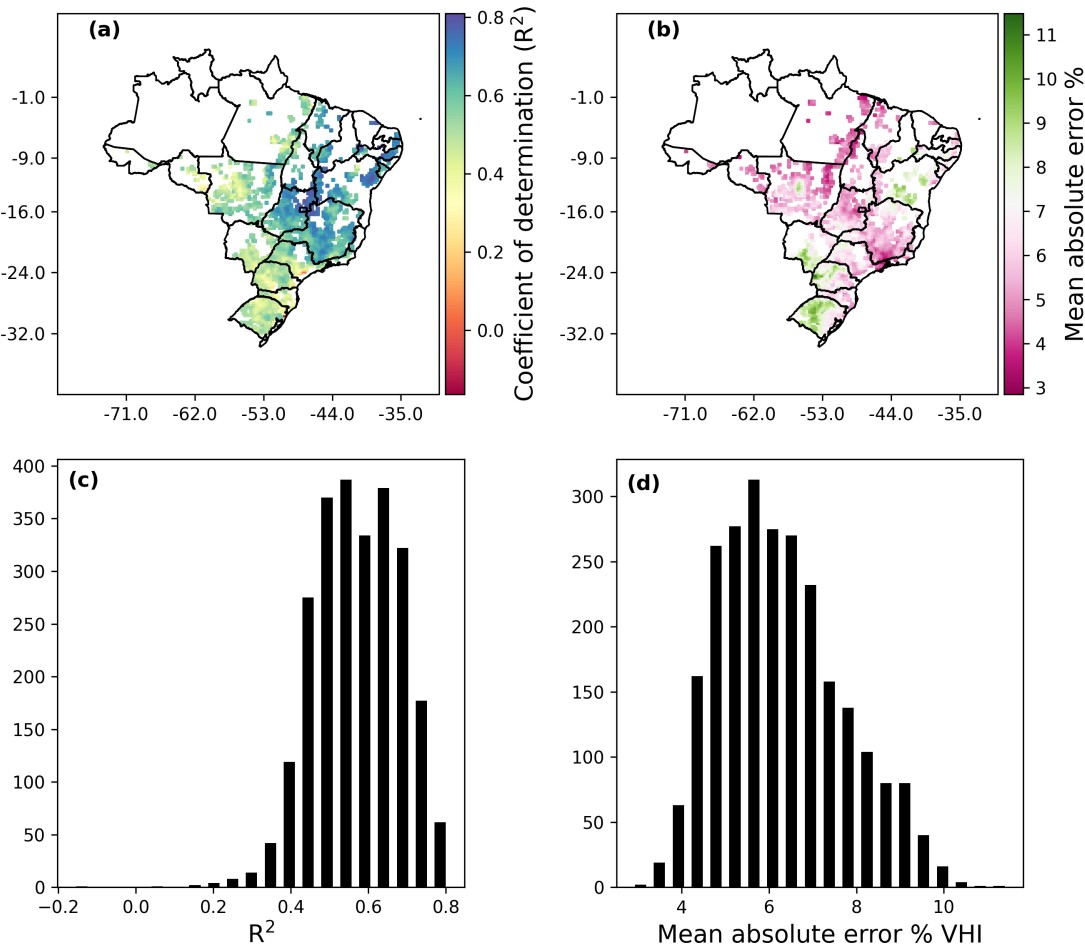

**Figure 7.** VHI forecasting model performance for the best model across the spatial domain in Brazil, showing the $R^2$ score and mean absolute error for each grid cell location.

Across months, VHI forecasts show little difference in the distribution of mean absolute error (Figure 8). However, coefficient of determination values can differ much more between months. Panel (a) of Figure 8 shows that January, February, September and October are typically the most difficult months to predict. Because $R^2$ differs more than mean absolute error, this indicates that variability is more poorly captured in these months rather than a particular overall bias relating to the mis-characterization of the seasonal cycle of VHI.


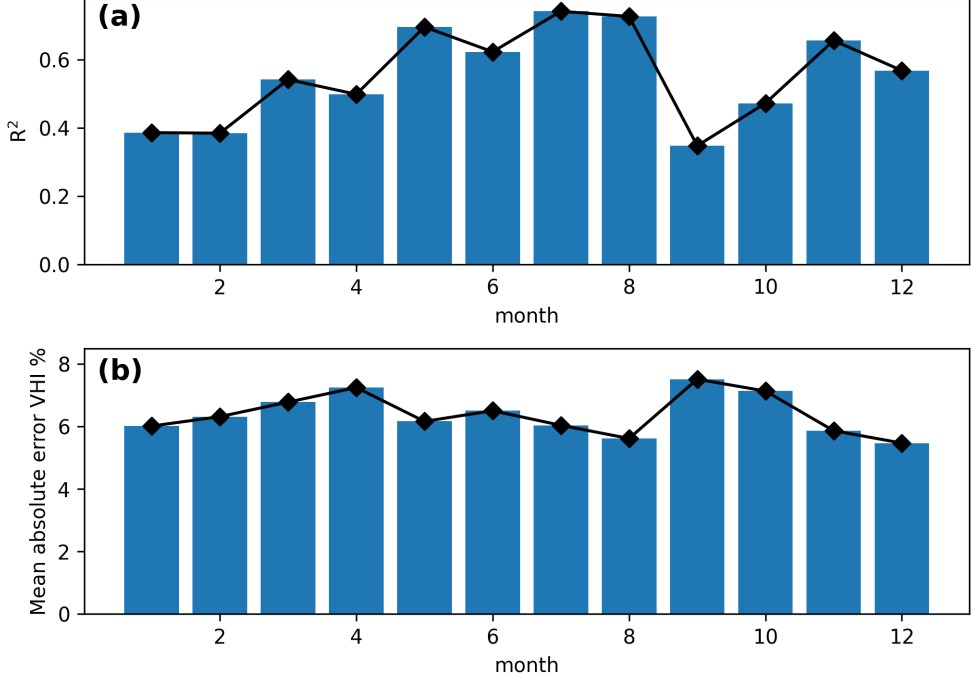

**Figure 8.** VHI forecasting model performance for the best model summarised as an average for each month, showing the $R^2$ score and mean absolute error per month.

Further subsequent months after 1 month into the future were assessed to determine how model performance reduces for increased lag times. Figure 9 shows how model coefficient of determination reduces from a median of 0.63 to 0.24 then 0.09 when increasing the forecast lag time from 1 to 2 then 3 months.


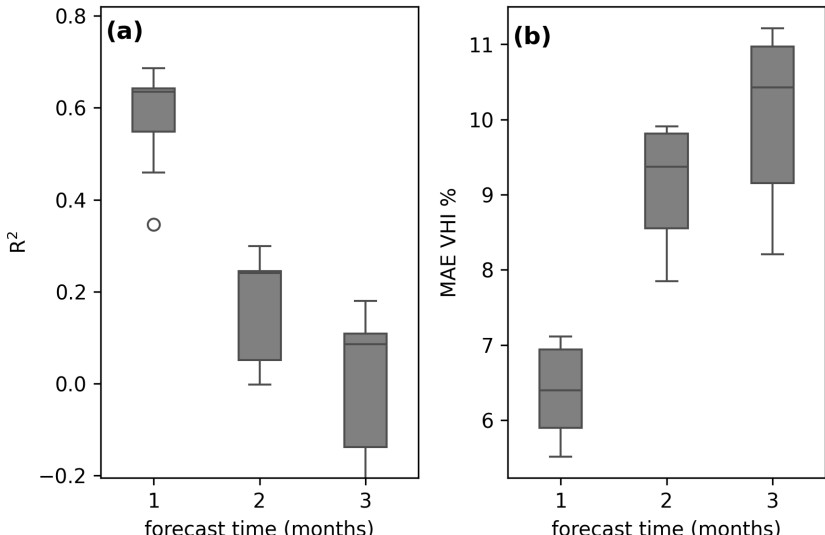

**Figure 9.** VHI forecasting model performance for the best model with comparison against increased lag times of 2 and 3 months into the future.

### 3.4 Effects of Southern oscillation index

Brazilian agriculture can be significantly affected by changes in the El Niño southern oscillation (ENSO). ENSO will have
315 different effects in different regions of Brazil (Cirino et al., 2015; Júnior et al., 2020). Here, model performance metrics are split into El Niño and La Niña evaluation periods. Figure 10 shows how spatial trends in model performance can be affected by ENSO. For El Niño periods, model $R^2$ significantly reduces for the central region (particularly central Mato Grosso state), and there is a broader trend of decreases in model $R^2$ values across the South. This trend is also shown for mean absolute error. Generally, La Niña periods are forecast better than El Niño periods.


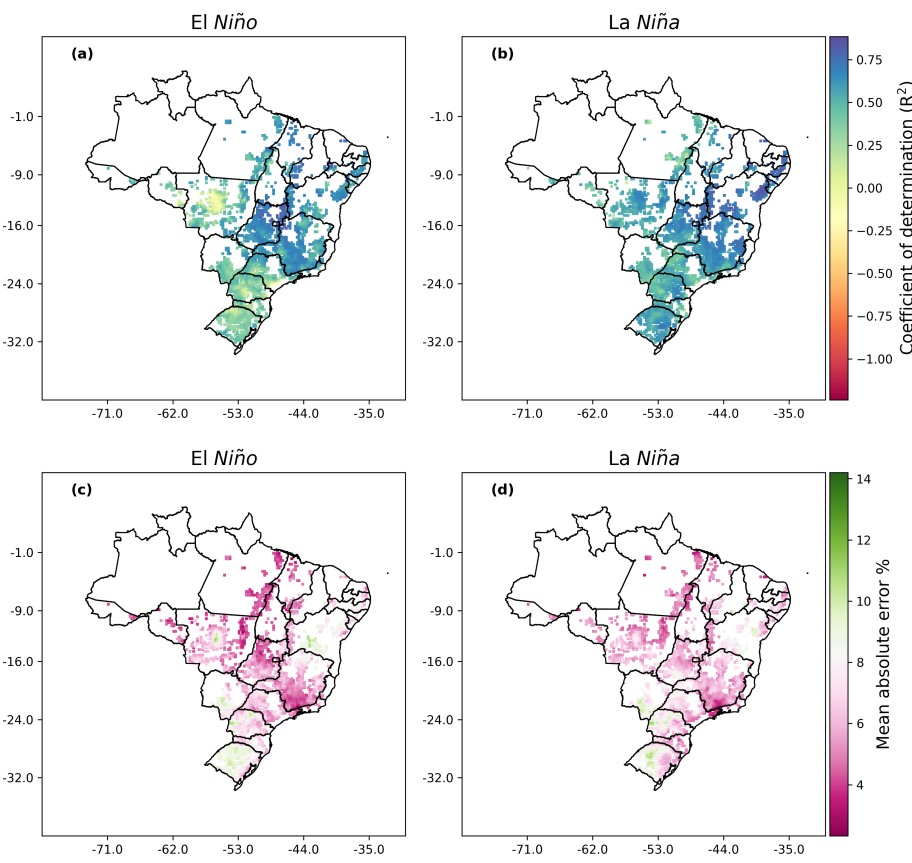

**Figure 10.** VHI forecasting model performance against Positive and negative modes of the southern oscillation index, negative SOI is associated with El Niño, and positive SOI is associated with La Niña weather events.

Figure 11 shows how the effects of the ENSO can lead to either under or over-prediction of VHI depending on location and ENSO mode. Of note is the over-prediction of VHI in central Mato Grosso in El Niño periods and under-prediction in the south. Generally, model performance is less affected by La Niña than El Niño.


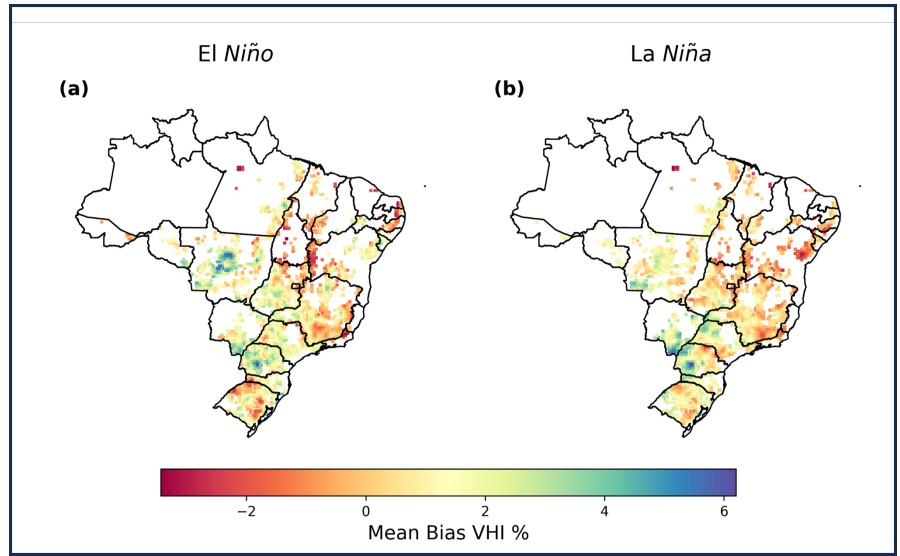

**Figure 11.** Bias in VHI model forecasts for El Niño (a) and La Niña weather (b) weather events.

## 3.5 Predicting onset of drought impacts on VHI

It is also important for models to be able to forecast when drought impact may reduce VHI below the typically used alert
threshold of 40% (Zeng et al., 2023; Kogan, 1995; Masitoh and Rusydi, 2019). The metrics described in section 2.6 are used
here to determine the performance of the best model for forecasting if VHI may fall below 40% in the following month.
Typically, model precision is greater than recall, meaning that there is a bias towards over prediction of values above 40%
rather than over-prediction of values below 40%. This is to be expected given the distribution of VHI values results in more
values above this value.

Figure 12 shows overall recall and precision (a) and when separated to El Niño and La Niña weather weather events (b). The
El Niño affects model performance by increasing the range of precision and recall values, increasing the number of locations
which are poorly predicted.


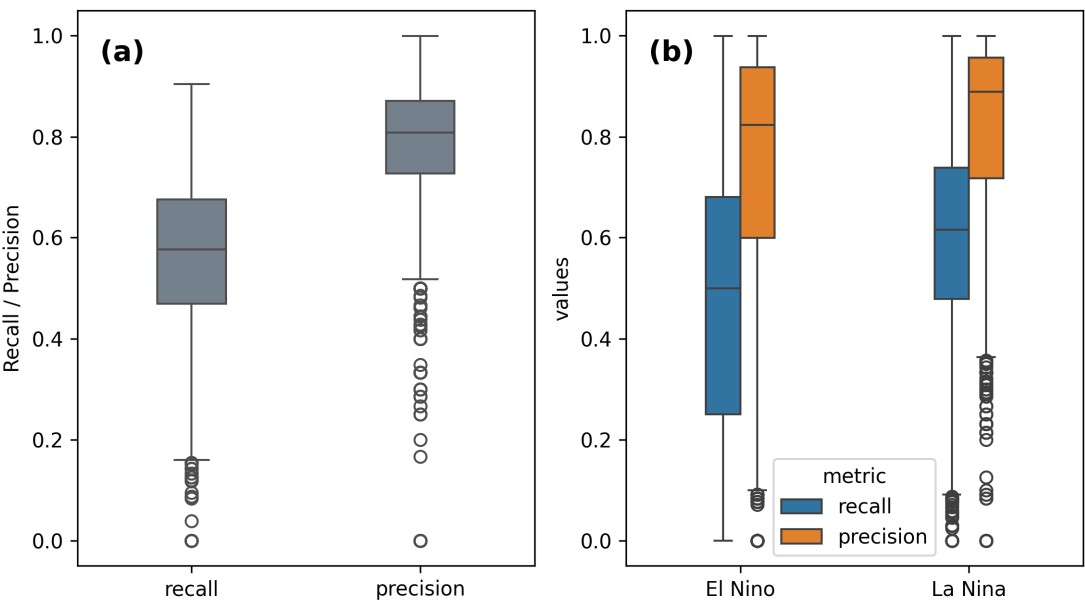

**Figure 12.** Recall and precision box plots for both all data combined (a) and for El Niño and La Niña weather events separately (b). Each point is a grid cell.

Figure 13 shows the spatial pattern of recall and precision. Generally, recall is lower than precision although there is no clear spatial trend. Lowest recall tends to be in coastal areas.


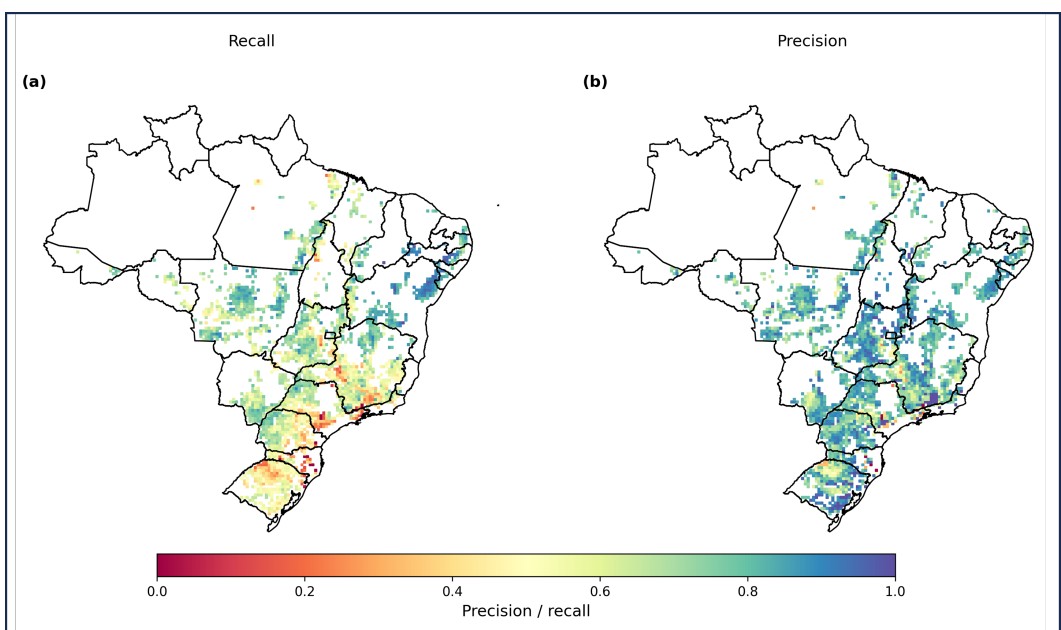

**Figure 13.** Maps to display recall (a) and precision (b) for each grid cell.

## 3.6 Feature importance

With meteorological data, ML model feature importance can be obscured due to high correlations between input variables. Standard techniques to assess feature importance such as permutation importance do not account for correlations between inputs and so can underestimate the importance of correlated features (Molnar, 2022). Figure 14 shows Spearman rank correlations between each of the input features. Highest correlations between features are between SPEI2 and 3, and SPI2 and 3 respectively, both have a Spearman-rank correlation of above 0.8. Secondly, t2m and longrad are also highly correlated (0.79). To consider shared information between variables, Figure 17 shows hierarchical clustering between highly correlated variables.

A further variable was also included in initial tests (and in Figure 14) which was the Southern oscillation index (SOI). SOI provides an indicator of the mode of the ENSO which can show whether El Niño or La Niña conditions are likely to occur. The southern oscillation index was however shown to provide little information gain in preliminary model tests and adversely affected model performance in some instances. Therefore, this index was not included in the model results presented in this paper.

To coincide with standard measures of feature importance, correlations are measured between the strength of correlation between input and output variables with model performance. This is shown for SPI and SPEI indices in Figure 15 and for temperature and VHI in Figure 16. Here we show the relationship between the strength of correlation between input variables and observed VHI, and model performance measured by coefficient of determination.

Figure 15 shows that model performance is more highly correlated with longer accumulation periods of SPI and SPEI. Furthermore, SPEI likely has a greater effect on model performance than SPI. Figure 16 indicates that 2 metre temperature





may be a stronger variable to use to capture the effects of temperature on VHI than other similar but correlated variables such as incoming longwave radiation and potential evaporation.



**Figure 14.** Spearman-rank correlation between proposed predictor variables and the target variable (VHI), lag-1 denotes a lag time of 1 month relative to time step of VHI.

**Figure 15.** Pearson correlation coefficient between indices SPEI and SPI (1-3) and VHI against model prediction performance measured by coefficient of determination ($R^2$). A line of best fit is plotted in blue for each panel with R value in the bottom right. Each point represents the modelled $R^2$ at a single grid cell with corresponding SPEI/SPI and VHI relationship.


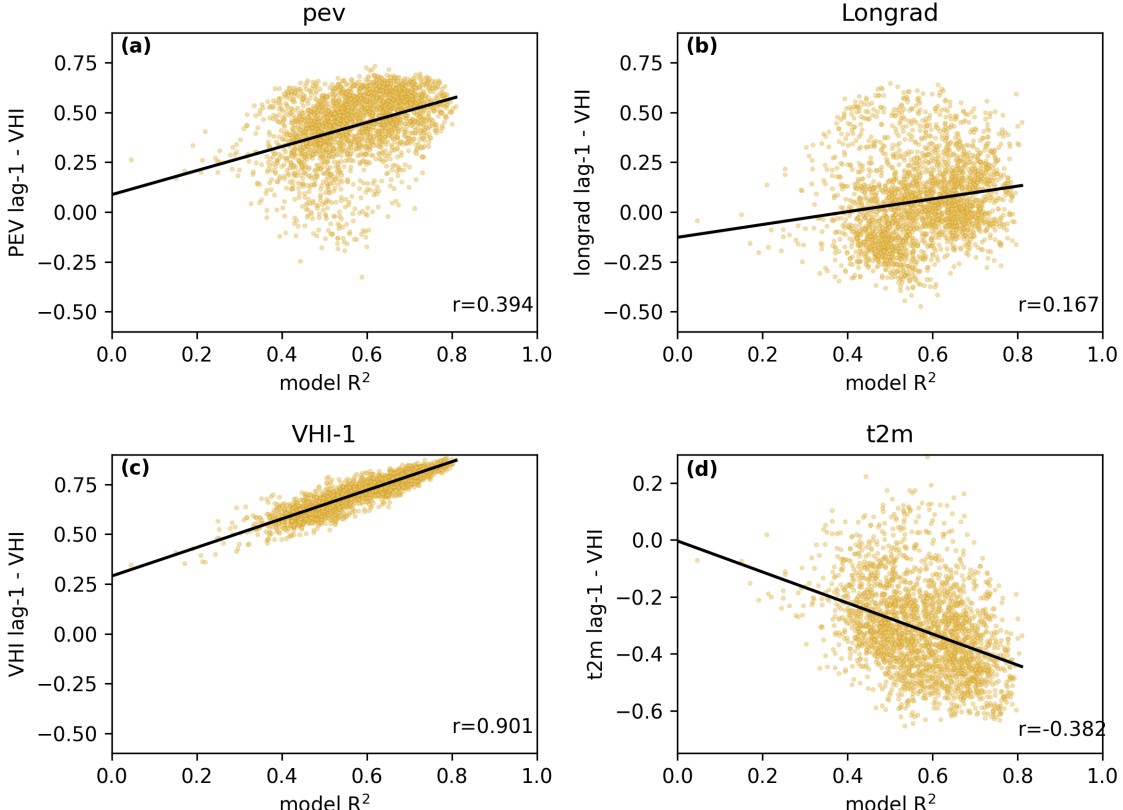

**Figure 16.** Pearson correlation coefficient between temperature effects and VHI against model prediction performance measured by coefficient of determination ($R^2$) as well as VHI autocorrelation. A line of best fit is plotted in blue for each panel with R value in the bottom right. Each point represents the modelled $R^2$ at a single grid cell with corresponding variable and VHI relationship.

As an additional analysis, Shapley values were obtained for the gradient boosting method used in this study. Figure 17 displays Shapley values obtained. Values indicate, similarly to the above analysis, that VHI obtained for the previous month is a strongly contributor in determination of VHI for the next month. The month variable is likely also highly influential as it is used as a proxy for seasonality. Other variables which are highly correlated (such as SPEI2 and SPEI3) may have lower contributions due to correlations between variables. To compute Shapley values, one trained model is required. Therefore, we used a random set of five years to evaluate model performance and the rest of the data was used for training.


**Figure 17.** Shapley values obtained for each of the model input variables used for the gradient boosting model.

## 3.7 Spatial scale effects on model performance

Here results of the clustering analysis are presented. model performance generally decreases with less data although results can vary depending on years used. Figure 18 shows the spatial distribution of clusters when specifying 1-10 cluster centres.

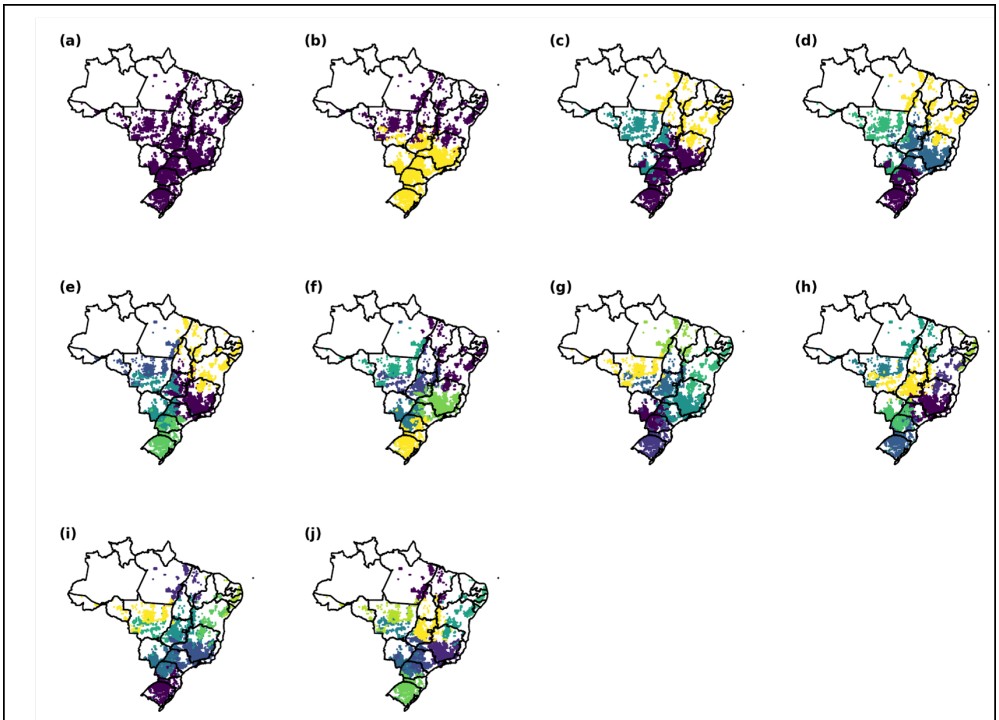

**Figure 18.** Spatial distribution of cluster sets used to determine effects of spatial sample size. Annotations (a)-(j) denote 1 to 10 clusters.

Figure 18 shows that with 2 clusters data is split between north and south, with 3 and 4 clusters, distinctions are made
between the northeast, central regions and southern Brazil. Increases in the number of clusters beyond this number is likely too
many splits in the data set, however clustering continued to demonstrate the effect on model performance.

Effects on model performance are shown in Figure 19. Greatest difference between sets of clusters are found using the model
coefficient of determination metric. Average model $R^2$ generally decreases from the most amount of data and spatial variance.
When training and evaluating more than once, results can differ depending on the years used. However, across multiple models
trained, typically model performance is greatest for the most amount of data.


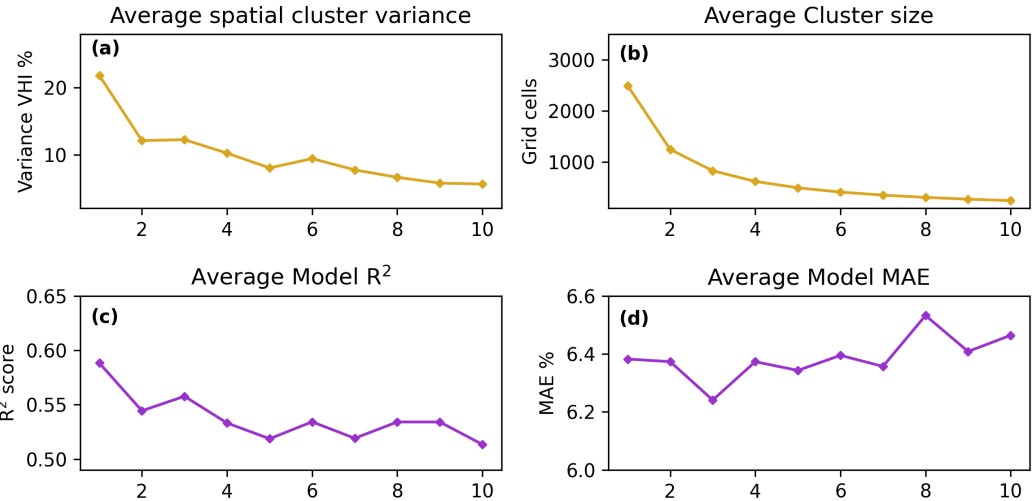

**Figure 19.** (a) Average within cluster spatial variance, (b) average number of locations per cluster, (c) Average $R^2$ score per cluster, (d) Average MAE score per cluster.

## 4 Discussion

Model performance depends upon the relationship between soil moisture, SPEI, SPI and VHI across Brazil, as well as the temporal autocorrelation of VHI. VHI in Southern Brazil (where rainfall is generally higher) is generally more difficult to forecast. For this reason, forecasting models presented here are most appropriate to be used in primarily moisture driven regions (with most useful results for the north east). The results presented here are of great significance for drought monitoring and forecasting efforts in Brazil and for others who may use studies such as this to inform other drought monitoring and forecasting work for other countries and regions. Here we show that machine learning is capable of accurately forecasting the spatio-temporal variability in VHI across Brazil, and can also determine when VHI values are likely to fall below the 40% drought threshold. Gradient boosting methods are an excellent all round method to use for both these evaluation metrics. Model performance is affected by El Niño events in the south and central Mato Grosso. Although machine learning is able to forecast when VHI will fall below 40%, typically, model precision is greater than recall. This means that the model is more biased towards forecasting VHI values above 40% than below which is as expected given the distribution of VHI values.

### 4.1 Regional variability in VHI

Vegetation health index variability is greatest in the northeast semi-arid region of Brazil. In this region, VHI is more greatly driven by rainfall and subsequent moisture effects than any other region in Brazil. This makes the northeast the most easily forecast region. In The south and mid-west region (particularly Mato Grosso State) This trend results in poorer model performance. Subsequently, temperature effects are greater drivers of VHI in these regions. This is likely due to the regional effects





of limiting factors, which limit the growth of crops and vegetation and are known to vary spatially with varying climate (Sacks et al., 2010).

## 4.2 Spatial heterogeneity and temporal autocorrelations

Temporal autocorrelations across space indicate high monthly autocorrelation for VHI and RZSM. These temporal autocorrelations help to improve the ability to forecast VHI on monthly timescales. VHI temporal autocorrelation is highest in the northeast, This is a decisive factor contributing to greater model performance in this region. RZSM generally has a high monthly autocorrelation, however this is reduced in the northeast. The semi-arid region typically has drier soils with reduced soil moisture capacity. This may be leading to a greater correlation with SPEI and SPI in this region in comparison to RZSM as this is not a principal driver of VHI.

## 4.3 How to build the most useful model for sub-seasonal VHI forecasting in Brazil

The results presented here provide key insight into the development of machine learning methods to forecast the effects of drought on vegetation health. This ranges from the indices and variables most useful, to the spatial scale which is most appropriate to train models to maximize the trade off between spatial variance and number of data points. Recommendations for how to build a forecasting model come in the form of three key factors: ML architectures, indices, and spatial scale. Accurate forecasting requires a method of appropriate complexity (Challinor et al., 2009). The appropriate level of complexity should strike a balance between model explanatory power and number of parameters to constrain. This study clearly indicates that linear methods such as multiple linear regression lack the explanatory power to effectively forecast forthcoming drought impacts and trends in VHI. Conversely, some methods may be too complex, in this circumstance, gradient boosting methods more consistently outperformed the artificial neural network. Neural networks contain a large number of parameters, which ultimately require more data to be adequately constrained, this can cause model training to be of far greater challenge.

Choice of climate indices and variables are also a key question when building a forecasting model. In Brazil, the wide range of biomes across the country can mean that the influence of certain indices such as SPEI and SPI may be of greater importance in some regions than others. Particularly, dry areas in the north east which are more effected by drought based indicators SPEI, SPI and RZSM. Furthermore, although SPEI may be more influential than SPI, it is more important that longer term indicators of 3 months are used above shorter 1 month accumulation periods. Of course, using the temporal autocorrelation in VHI is a key factor in determining model performance. Regions which have the greatest monthly VHI autocorrelation also are the most easily forecast. Temperature variables are more useful for the forecasting of VHI in south Brazil, where typically rainfall is higher and drought is less common in occurrence.

Model spatial scale refers to the trade off between including a wide spatial coverage and the models ability to generalize across the biomes and climates included in the dataset. Analysis in section 3.7 indicates that distinctions between climate zones are not necessary for best model performance. Further splitting of the dataset only reduces the overall amount of data and reduces model performance. This is a key result for future large scale modelling work across Brazil. Most importantly, this shows the excellent generalization power of machine learning, particularly over process based models used in agricultural





applications which will require more site specific data to achieve similar performance over such a broad range of climates (Angulo et al., 2013).

Here models were trained for VHI value forecasting and then the ability of the best model to determine onset of drought is found in section 3.5. this resulted in high precision with slightly lower recall, meaning that model bias is towards forecasting values of VHI above the 40% threshold. For the forecasting of drought onsets, a more effective method to train models may be to use a classification model with altered training data to over-sample VHI instances in which VHI is below 40%. There are many methods which can be used to improve data set balance and improve recall, such as ensemble based methods, over and under-sampling strategies, and synthetic minority oversampling methods (Chawla, 2010). However, in doing this, forecasting of VHI values would require a separate model.

## 4.4 Scope of methods analysed

Here we analyse machine learning methods including artificial neural networks, gradient boosting and random forest methods, nearest neighbour methods and linear regression methods. Among methods excluded include convolutional LSTM models as discussed by Kladny et al. (2024) as well as other deep learning methods such as an ensemble of temporal convolutional neural networks (Miller et al., 2023). These model frameworks were excluded from the methodology following the general principal of Occam's razor to evaluate simpler methods first before expanding the scope of the work to more complex methods with greater numbers of parameters in future work. Evaluating such methods in the region should be a priority for future work building from this study.

## 4.5 Future model developments for Brazil drought monitoring

To expand on the scope of this study, further work will focus on the application and assessment of machine learning architectures such as those described in the previous section (4.4). Such methods have been shown to improve vegetation health forecasts in other regions (Kladny et al., 2024; Miller et al., 2023) and so may also improve results here. Furthermore, improvements could be made to the forecasts of specific months key for agricultural production. Here, the best model trained can have variable model performance depending on month of assessment. A greater assessment of sampling methods or the targeted use of model ensembles may improve model the stability of model performance for key months. For many regions November - March of the next year can encompass a typical growing season (CONAB, 2022). Therefore, these months are of greater importance. For key growing season months such as time of planting or anthesis, it becomes more challenging to determine when the critical periods are for these dates given the lack of high spatial resolution planting data.

This work aims to inform future developments in drought monitoring for Brazilian agriculture at CEMADEN. Forecasting VHI would help to identify areas potentially affected by drought one month in the future. Currently, forecasting of next month's SPI is used to measure the potential impacts of drought, since rainfall anomalies are critical as a hazard. The forecast of VHI can bring information on potential impacts, since it reflects on the vegetation health. This information is essential for disaster preparedness and planning of future actions to support areas affected by drought. The identification of drought evolution can





inform decision makers in several agencies and levels of government on how to manage resources destined to alleviate drought impacts on agricultural activities.

## 455  5   Conclusions

This study addresses several questions important for building a drought impact forecasting framework for Brazil. Here we show the importance of different meteorological variables and indices for monthly VHI forecasts, a new baseline of model forecast performance across Brazil, and evaluation of model performance for key factors relevant for stakeholders in Brazil such as model performance as a function of ENSO mode, ability to predict onset of drought impacts, and best spatial coverage
for model training. Significantly, we show that machine learning methods are able to forecast monthly drought impacts on VHI and VHI trends across a very large scale containing multiple biomes with broad climate regimes. Future work should aim to build upon these results to further aid drought monitoring efforts with improvements to model performance through additional pre-processing techniques and further assessment of machine learning modelling frameworks.

*Code and data availability.*  Code is available upon request of the corresponding author.

## 465  Appendix A:  Hyperparameter optimizations

Here we show results from the optimization of certain hyperparameters for the models investigated in this work. Hyperparameters are global parameters which affect the learning process rather than the model itself. Hyperparameters can include the learning rate of a neural network, the number of neighbours to use in the k-nearest neighbours algorithm, or the number of decision tree estimators present within a random forest model or gradient boosting machine. We undertook minimal hyperpa-
470 rameter optimization. We use the coefficient of determination to optimize hyperparameters across cross validation folds. For gradient boosting and random forest models, we optimize the number of estimators which comprise the model. We found that above a certain threshold value (typically 2-10) the number of estimators which achieved the best results can vary if repeating optimization. For the K-nearest neighbors algorithm, the number of neighbors was varied between 5 and 1000, 500 neighbors was found as the optimum value. For Ridge regression, the regularization parameter, ($\alpha$) was optimized however results did
not improve above those of the default value (1). The neural network was optimized by varying the number of neurons in each hidden layer, and the number of epochs, which is the number of iterations through the dataset when training. Through optimization we determined 20 epochs with 12 neurons in each layer. We kept the number of hidden layers as small as possible (1 hidden layer) to avoid over-parameterization. Model results with these optimized parameters are found in Figure A1.


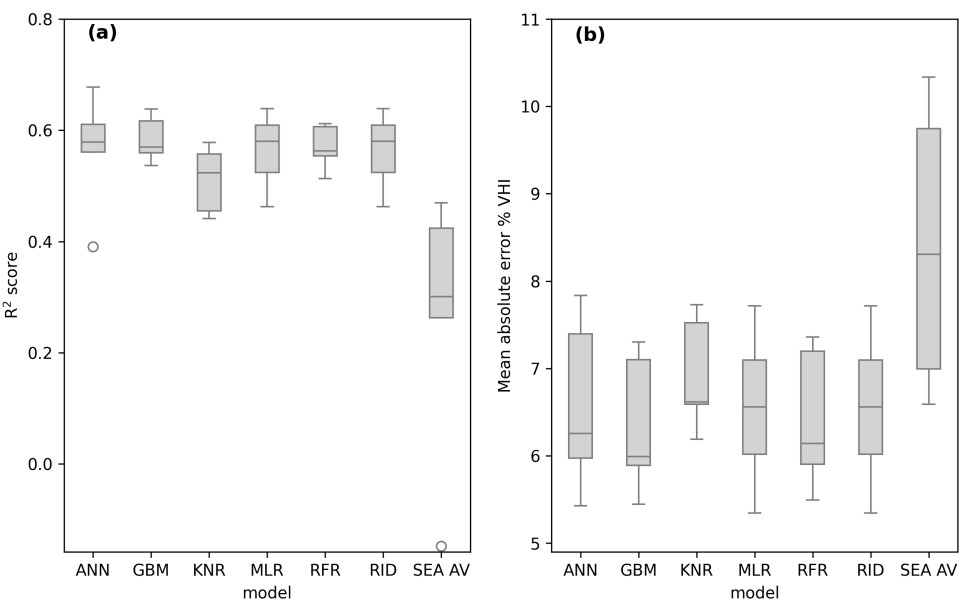

**Figure A1.** VHI forecasting model performance across the hold out evaluation data set for each of the initially selected models.

Some models achieved slightly better performance with optimization such as KNR. However, more data generally resulted
in better model performance rather than optimized hyperparameters. Gradient boosting (GBM) is the best performing model
regardless of hyperparameter optimization.

*Author contributions.* Joseph W Gallear wrote the text, developed the code and Figures and generated the ideas for the methods, Marcelo
Zeri wrote some of the text for the introduction, methods and discussion, provided data, generated ideas for the methodological process,
and provided feedback on preliminary results and discussion. Marcelo Valadares Galdos generated ideas for the methodology and provided
feedback on the text of the paper and preliminary results and discussion. Andrew Hartley generated ideas for the methodology and provided
feedback on preliminary results and discussion.

*Competing interests.* The authors declare that they have no conflict of interest.

*Acknowledgements.* This work and its contributors (Joseph W Gallear, Marcelo Valadares Galdos, Marcelo Zeri, Andrew Hartley) were
funded by the Met Office Climate Science for Service Partnership (CSSP) Brazil project which is supported by the Department for Science,
Innovation & Technology (DSIT).





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
