# Peer review of "Application of machine learning to forecast agricultural drought impacts for large scale sub-seasonal drought monitoring in Brazil"

_Natural Hazards and Earth System Sciences, 2024_

## Author Response (AR1)

| Number | Comment | Adjustments | Relevant sections / page numbers |
|---|---|---|---|
| *1* | *It uses as predictors climate variables, soil moisture, and drought indices such as SPI and SPEI at multiple time scales. The article lacks the use of substantial and proper references about the matter* | Included new references for the calculation of SPEI and SPI to the methods section | Pages 5-6, sections |
| *2* | *The article does not state a clear definition of the objectives, which could lead to a better structure for the manuscript. Due to this, it was hard to follow and understand the storyline* | Included new objectives list at the end of the introduction | Page 3, section 2.1 |
| *3* | *The word "impact" in the title induces me to think about the consequences of drought (socio-economic, human, etc.). The title should be improved* | Changed title to remove the word impact | Title |
| *4* | *The introduction is not easy to follow; it doesn't have a clear scientific meaning, has vague sentences, and does not have a proper order of ideas. It needs some improvements to make it scientifically sound. For example, the paragraph talking about drought impacts on Brazil should be moved up before the paragraph talking about drought monitoring (L35)* | Restructured introduction entirely. 3 paragraph structure now follows as Introduction to drought, then discussion of the topic of drought monitoring, then specifically drought monitoring in Brazil | Pages 1-3, section 2 |
| *5* | *Also, it should be presented with some numeric figures of the real impact rather than solely indicate where it has impacted* | Included cited example of drought impact causing economic losses and figures relating drought to inflation and food price increases in Brazil | Page 2 section 2 |
| *6* | *The paragraph describing previous works using machine learning lacks robustness; it should not only describe what types of methods have been used but also describe what results these works have had* | We have included new text to describe the results of some relevant studies to demonstrate the value of the methods used in this work | Page 2, section 2 |
| *7* | *The definition of the objectives is vague; the authors should go directly to the scientific aims of the work rather than deviate toward the potential benefits of the results or come back to defining the importance of the indices. Defining* | Included new objectives list at the end of the introduction | Page 3, section 2.1 |

| | | | |
|---|---|---|---|
| | *two or three clear objectives that will lead the work is preferred.* | | |
| *8* | *It is needed to provide a better description and justification for the use of the crop grid dataset (Tang et. al, 2023) instead of just presenting the reference* | Included the justification of using the crop grids dataset under study area "The crop grids dataset was chosen because it is the newest dataset found with estimates of crop specific growing area for maize and soybean in Brazil." | Page 4, section 3.1 |
| *9* | *In the sentence (L130), it says, "... NDVI which is in turn converted to TCI, VCI, and the vegetation health index (VHI)," which implies that the TCI is derived from NDVI when in fact it is derived from the thermal bands* | This sentence has been removed | Page 5, section 3.2.1 |
| *10* | *Please add a table for a better comprehension of all the satellite data used in this work and its characteristics* | We have included information on VHI in the already existing predictor variables table | Page 8 section 3.3 |
| *11* | *There is too much description of the data used, but it was not calculated in this work. I believe it should be good to reduce this and focus on what was made in this work, i.e., forecasting by machine learning. The acronym for the method used is not described here* | Removed detail from descriptions of VHI, and RZSM data | Pages 5-7, section 3.2.1 |
| *12* | *I think it is important to somehow analyze the spatial cross-validation. But, as it is currently written, it is not clear to me what the purpose of the spatial clustering analysis is. It needs further clarification regarding its link with the rest of the methodology. Also, authors should not cite figures from results in the methods section. The cluster allows for the splitting of the data (training and testing), and it affects the regression (forecasting) and classification (onset of drought)* | The section on spatial clustering has been removed from the paper. This decision was made for several reasons including clarity and flow of the story of the paper. | n/a |
| *13* | *There is an excessive number of figures that can be reduced. For example, Figs. 14, 15, and 16 could be reduced to one and perhaps a table summarizing the results* | Removed Figures on temporal autocorrelations and variability, Combined Figures on future months and monthly | n/a |

| | | model performance and moved to the appendix

Combined Figures which describe recall and precision into one.

Reduced number of Figures in the main text to 8 | |
|---|---|---|---|
| 14 | *The study area includes crops of soybean and maize; it would be good to know how the ML model performs per crop type* | Some discussion has been added to explain why soybean and maize growing areas are used together rather than separately | Page 29, section 5.4 |
| 15 | *The quality of Fig. 17 is poor.* | This Figure was removed | |
| 16 | *The conclusion is too general and does not specifically state what the main results are. What variables were the best predictors for VHI? What machine-learning methods achieve the best performance? What is the main contribution to the drought research in the article?* | Rewrote the conclusions to be clearer and provide the significance of the work for both Brazil and broader agricultural drought forecasting | Pages 29-30, section 6 |
| | Reviewer 2 | | |
| 17 | *There are too many paragraphs in the introduction. I suggest to rewrite the introduction with 3 paragraphs, highlighting the basic content of the research field in the first paragraph. Then review the research progress of the literature in the second paragraph, and in the third paragraph, analyze the limitations of past research and clarify the innovation of your own research* | Restructured introduction entirely. 3 paragraph structure now follows as Introduction to drought, then discussion of the topic of drought monitoring, then specifically drought monitoring in Brazil | Pages 1-3, section 2 |
| 18 | *The importance of ML compared to other methods such as statistical, probabilistic, and time series modeling for drought monitoring and drought forecasting is missing in the introduction section. I would suggest to add this in the introduction section* | Explained the advantages of ML methods as opposed to statistical approaches in the introduction, provided a reference to paper which compares a statistical and machine learning approach for crop yield prediction | Page 2 section 2 |
| 19 | *The description under section 2 (page 3; line 85, 90, and 95) is not so much important. Please delete these lines* | These lines have been deleted | Section 2 |

| 20 | Rewrite the study area highlighting the key geographic features, climate, and physiography of the study area. Please omit the first three line of the study area | Deleted first 3 lines and included a paragraph on the different climates and biomes of Brazil | Pages 3-4, section 3.1 |
|---|---|---|---|
| 21 | The author used 1,2- and 3-month SPI. Why did the author not use the SPI 6? SPI 6 indicates the seasonality of agricultural drought | Provided a justification for the exclusion of accumulation periods above 3 months | Page 6, section 3.2.3 |
| 22 | Why was only precipitation used as a predictor variable? Was it average precipitation or total precipitation? I think using only precipitation does not make any sense as the author used SPI and SPEI index, which is the form of precipitation-based drought index. In this regard, I would suggest adding precipitation anomaly index (PAI) as a predictor variable instead of only using precipitation. | Made it clearer that precipitation is total monthly precipitation. Provided a justification for the use of total monthly precipitation | Page 7, section 3.2.6 |
| 23 | In case of the machine learning model what amount of data was used for training, validation and testing of the model? I mean, how was the model built? How was it calibrated? The most important parameters and the choice of values for the model were not explained sufficiently much more explanation needed | Section labelled Cross validation and training procedure now more clearly describes the training, optimization and evaluation of the models | Page 8, section 3.5 |
| 24 | What does "SEA AV" mean for? What kind of model was it? What is the utility of using "SEA AV" model? | Provided a paragraph in the methods section to describe the purpose and meaning of the SEA AV model | Page 8, section 3.4 |
| 25 | Page-16 (line 300): Please close the first bracket for "(Figure 7 | Further proof reads of document | n/a |
| 26 | The author discussed only the forecasting performance of various machine learning models. But I did not see any forecasted results of VHI by the machine learning model, which performed better compared to other models. It is very important to add results of forecasted VHI by the best machine learning model. | Changed text to make more clear that GBM model is the best and is used exclusively for some analysis | Pages 13-14 section 4.2 |
| 27 | Conclusion can be improved by highlighting the innovation content of the paper, future research direction, and recommendation for policy formulation | Rewrote the conclusions to be clearer and provide the significance of the work for both Brazil and broader | Pages 29-30, section 6 |

| | | agricultural drought forecasting | |
|---|---|---|---|

---

## Author Response (AR2)

Thank you for taking the time to provide comment and feedback on our manuscript. Please see below for a list of changes made and responses to each of the comments made in the review.

| | Comment | Adjustments | Section |
|---|---|---|---|
| | **Introduction** | | |
| 1 | The first paragraph should be for agricultural drought monitoring. The second paragraph for machine learning models used to forecast agricultural droughts. | Thanks for the feedback. The structure of the introduction is based on the comments made by reviewer 2 in the previous review. (see bullet point 1 of https://doi.org/10.5194/nhess-2024-60-RC2)

We believe it would be wrong to make further changes to the introduction which would change it from which was originally requested by the second reviewer | n/a |
| 2 | The second paragraph should begin with "Machine learning has been shown to outperform...", all the previous text should be in the first paragraph. | Begun second paragraph with this statement, prior text was moved to first paragraph | Page 2 section 1 |
| 3 | L20 "Drought are defined as an extended period in which a water deficit occurs, usually because precipitation is less than average resulting in water scarcity (Cunha et al., 2019)."

What type of drought are you defining? In your manuscript, you are using VHI, which depends on temperature and vegetation. Thus, VHI is a drought index that does not depend on precipitation. A better definition of drought types is needed. | Specified that the initial definition of drought provided is that of meteorological drought

Included a definition of agricultural drought at the end of the first paragraph | End of 1st paragraph of introduction |
| 4 | L20 "Agricultural drought can have significant socio-economic impacts because they impact food security."

However, you have not provided a definition for agricultural drought. | Included a definition of agricultural drought at the end of the first paragraph | End of 1st paragraph of introduction |
| 5 | L25 ". This is because VHI, derived from AVHRR (Advanced very high resolution radiometer) data, responds cumulatively and quickly to changes in vegetation greenness."

Remove the word "derived". | Removed word derived | Paragraph 2 of introduction |
| 6 | L30 "Drought monitoring using vegetation indices such as VHI or | We have included a definition of VHI, VCI and TCI to clear up any confusion | Introduction paragraph 1 |

| | | | |
|---|---|---|---|
| | NDVI (Normalized difference vegetation index) or VCI (Vegetation condition index) has been developed in several locations using satellite imagery from products such as MODIS, and NOAA STAR (Sadiq et al., 2023; Kloos et al., 2021). "

This sentence is confusing because VHI = a*VCI+(1-a)*TCI, and VCI uses NDVI and TCI uses BT or LST | | |
| 7 | L30 "VHI is reported to improve on NDVI based monitoring as it provides a measure of vegetation condition relative to long term change (West et al., 2019)"

VHI uses VCI, which in turn uses NDVI. Therefore, I am uncertain whether the authors have a thorough understanding of the VHI drought index at this point. Before these sentences, at least, we need a clear definition of VHI. | We have reworded this sentence to make clear that VHI is an improvement on using NDVI but is still based on NDVI.

We have also included a definition of VHI, TCI and VCI | Introduction paragraph 1 |
| 8 | L55 " For example, in 2020 drought in Rio Grande do Sul was estimated to have cost R$ 36 billion in losses representing 7.36% of the states GDP (CNA, 2020). "

For a wider audience, you should express numerical figures in dollars. | Added US dollars in brackets | Section 1 page 3 |
| | **Methods** | | |
| 9 | The study area should come first; the previous introductory paragraph is unnecessary. | Introductory methods paragraph was removed | Section 2 |
| 10 | L95 "...up of 9 different Köppen-Geiger climate zones from semi-arid in the northeast,.."

For figures up to ten, should be preferred to use words rather than numbers. | 9 converted to words | Section 2 |
| 11 | L110 "The data was filtered using harvested areas from the crop grids dataset (Tang et al., 2023"

Which data are you talking about? | Changed text to specify that all input and output data is filtered using harvested areas from crop grids | Section 2 paragraph 2 |
| 12 | L160 "SPI indicators with longer accumulation periods were not | Included a new appendix E which provides a greater justification for not using SPEI 6 and 12, | Appendix E |

| | | | |
|---|---|---|---|
| | tested because such accumulation periods would be longer than the growth periods of maize and soybean. SPI is a widely used index recommended by the world meteorological organisation (WMO). It is also used for operational drought monitoring at CEMADEN (Cunha et al., 2019)"

The justification is insufficient to warrant the avoidance of using higher time scales. It will be worth analyzing up to a 12-month timescale. | furthermore, I have also completed a correlation analysis to show that VHI is more strongly correlated with SPEI 2 and 3 than 6 and 12. | |
| 13 | Why are soil moisture, total precipitation, and ERA5 included in the "Drought indices" section? These are not drought indices, but rather predictors. | Changed title of section to be input variables & drought indices | Section 2.2 |
| 14 | L200 "Where spatial resolution has increased (spatial up sampling) this is calculated using a k-nearest neighbours algorithm

This was nearest neighbor (k=1) or k-nearest neighbors? If the latter is the case, what was the value of k? Most GIS tools use nearest neighbors for resampling continuous data. | Provided details on the value of K used for the spatial subsampling | Section 2.3 page 7 |
| 15 | L210 "user. random forest constructs a specified number of trees and then averages the result of each individual tree. Different trees

After the dot (.) it should start with a capital letter? | This has been corrected as part of a further proof read | Throughout |
| | Results | | |
| 16 | In my previous review, I added the comment, "The study area includes crops of soybean and maize; it would be good to know how the ML model performs per crop type." But it wasn't addressed by the authors. | This comment was previously addressed in the reviewer comments response table point 14. We also included a justification for using maize and soybean growing area together rather than separately

Included new appendix D which shows negligible differences between model performance when training the model on maize and soybean separately. | Appendix D |

| 17 | L320 "Feature importance"

You should choose to use "feature" or "predictor", but not both, because it is confusing. | Replaced all instances of the word "feature" with "input variable" | Throughout |
|---|---|---|---|
| 18 | In machine learning, you must calculate variable importance independently. For instance, a random forest utilizes out-of-bag (OOB) bootstrapping to estimate the importance of each feature (aka predictors). Typically, a single plot displays the importance of each feature, allowing you to compare them and determine which feature significantly influences the model's performance. | We opted for 2 complementary methods to calculate variable importance. The first is to determine the correlations between model performance and skill, (shown in Figures 7 and 8) and the other is to calculate Shapley values. The plot showing Shapley values was originally in the paper but it was described as 'poor quality' in the previous review. Without further explanation of what this meant the decision was taken to remove the Figure. For this latest version of the paper the Shapley values plot have been added back to the paper to provide a measure of variable importance in which the importance of each variable is calculated independently. | Section 3.5 |
| | **Discussion** | | |
| 19 | The study lacks a comparison of the forecast model's performance against other comparable studies to provide context. | Added a new section in the discussion which describes some results of other work which forecast NDVI, VCI and other vegetation indices at various timescales and addresses how the results of our study fit within the wider context. | Section 4.5 |